# Light and Ethephon Overcoming Seed Dormancy in Friar’s Crown (*Melocactus zehntneri*, Cactaceae), a Brazilian Cactus

**DOI:** 10.3390/plants12244127

**Published:** 2023-12-11

**Authors:** Mariana Freitas Campos Magnani, Jean Carlos Cardoso

**Affiliations:** 1Research Group in Physiology Applied to Agriculture, Center of Agricultural Sciences, Federal University of São Carlos, Rodovia Anhanguera, km 174, Araras 13600-970, SP, Brazil; 2Laboratory of Plant Physiology and Tissue Culture, Department of Biotechnology Plant and Animal Production, Center of Agricultural Sciences, Federal University of São Carlos, Rodovia Anhanguera, km 174, Araras 13600-970, SP, Brazil; jeancardoso@ufscar.br

**Keywords:** cactus, dormancy breaking, germination, light, phytoregulators

## Abstract

Seed germination in *Melocactus* and other cactus species is hampered by dormancy. However, most studies failed to achieve high seed-germination rates, suggesting a complex mechanism of dormancy in Cactaceae. Thus, the objective of this study was to evaluate whether factors such as light and phytoregulators overcome the dormancy in the seeds of the friar’s crown cactus (*Melocactus zehntneri*). Two consecutive experimental sets were designed: one with seed germination under filter paper conditions and different wavelengths and Photosynthetically Photon Flux Densities (PPFDs); and one in vitro experiment using a culture medium to evaluate the influence of different phytoregulators, such as gibberellic acid (GA_3_), benzylaminopurine (BAP) and ethephon (ET), both in the germination of seeds of *M. zehntneri*. Seeds of *M. zehntneri* are positive photoblastic. Red light and gradual increases in PPFD resulted in the highest germination rates (60.8–61.7%) and germination speed index (4.4–4.5). In vitro seeding in culture media increased the germination percentage to 76% in control without phytoregulators. Ethephon showed a major effect in releasing the germination of dormant seeds of *M. zehntneri*, totaling 98% of seeds germinated under in vitro conditions, while GA_3_ and BAP showed minor or no effect on germination. The present study resulted in an efficient in vitro technique for germination and a better understanding of cacti seed dormancy.

## 1. Introduction

The genus *Melocactus* comprises plants popularly known as ‘chapéu-de-frade’ or friar’s crown cacti, and Brazil is the country with the highest diversity, with 55% of the number of *Melocactus* species in the world occurring throughout the northeast region, and in parts of the north and southeast regions of Brazil [1].

The demand for *Melocactus* species in the commercial sector, especially for ornamentation and collecting purposes, has had remarkable growth in recent years, due to its peculiar aesthetics, ornamentation, and easy cultivation. However, it is difficult to be precise about the exact numbers of the trade of these plants since the illegal trade is difficult to quantify [2]. The fact is that many *Melocactus* populations have been drastically reduced [3]. Consequently, some *Melocactus* species are already under protection as a way of controlling the high extraction for trade [4,5]. The species *M. zehntneri* is classified as a least concern species, but its population has also shown a recent and constant decline and has recently been placed under protection [6,7]. In part, its population reduction can be attributed to two main factors: the continuous deforestation of the Caatinga Biome and the replacement of wild species with agriculture and livestock activities, and the recent increase of commercial exploitation resulting in the extraction of individuals for the production of food, traditional medicines, and animal fodder [8,9], but especially for ornamental purposes [10].

*Melocactus* displays a distinctive feature unique to the genus known as a cephalium at its apex. This characteristic occurs only when the cactus reaches reproductive maturity at approximately 10 years of age, mainly due to very slow vegetative growth (0.04–4.7 cm per year) [11]. The natural reproduction is exclusively by seeds due to limited vegetative propagation since they do not develop segmentations as observed in other cactus species [6]. In addition, seed germination in different species of *Melocactus* is hampered by dormancy. Different studies have reported the use of chemical scarification to increase germination rates and associated dormancy with the rigid seed coat. Also, phytoregulators, such as gibberellic acid (GA_3_), have been used to increase the germination rate in *Melocactus*. In *M. azureus*, the imbibition of seeds in water or gibberellic acid (GA_3_) for 2 h resulted in an increase of 14 and 20% seeds germinated, compared to only 3% germinated seeds in the control [12]. However, studies with other phytoregulators aiming to overcome seed dormancy are very limited. On the other hand, the key role of ethylene in the dormancy release of seeds of numerous plant species has been reported [13].

The use of in vitro germination has been proposed as an alternative to increase germination rates in *Melocactus*. Successful in vitro germination was reported for *M. glaucescens* (68.1%), *M. sergipensis* (64%), *M. zehntneri* (58.7%), and *M. violaceus* (59.3%) [14]. Additionally, previous results obtained by our group have demonstrated similar percentages of seeds germinated in Petri dishes (50–55% on average) for *M. zehntneri* [15]. Thus, the results obtained with different species of *Melocactus* showed that a substantial part of the seeds (40–80%) continues with some unknown type of dormancy, with no germination.

Thus, the present study aimed to test and determine the main factors affecting the germination of *M. zehntneri* seeds: the wavelength, light intensity, and phytoregulators, such as Gibberellic acid (GA_3_), Benzylmaninopurine (BAP), and Ethephon. From a practical point of view, the experiments aim to develop an efficient protocol for seed germination. This protocol serves two purposes: to enable in vitro conservation and to serve as a method for propagation of the *Melocactus* species, which could offer insight for conducting further studies on breaking seed dormancy in Cactaceae.

## 2. Results

### 2.1. Effects of Light Intensity and Wavelength on Germination

The percentage of seeds germinated using filter paper was affected by wavelength and PPFD, but the interaction between these two factors was non-significant. Thus, the effects of light wavelength (Figure 1A,B) and PPFD (Figure 1B,C) were analyzed separately. The highest germination percentage ranged from 60.8% to 61.7% and was reported in the white, red, and red/blue LEDs. The use of only blue LEDs resulted in a drastic reduction in seed germination to 37.5% (Figure 1A) and the lowest GSI values (Figure 1B). About the PPFD, the gradual increases of PPFDs of light resulted in a positive and significant correlation with the percentage of germinated seeds of *M. zehntneri* (Figure 1C). The GSIs values were also increased in the higher PPFDs (II and III), compared to the lowest PPFD used (Figure 1B).

The germination of the first seeds was reported 4 days after seeding (DAS) in the red/blue and white LEDs and at 7 DAS in the other wavelengths. The blue light resulted in the late beginning of germination at 9 DAS, with the lowest average germination speed (AGS) (Figure 1D). For the red, white, and red/blue LEDs, the maximum AGSs occurred between DAS 9 and 14 (Figure 1D). Increases in PPFD were those resulting in the highest values of AGS. The different wavelengths also affected the color of embryos after germination. In the red light, 100% germinated embryos had a light green color, related to chlorophyll biosynthesis; in the blue and white LEDs, the red–purple color was predominant in the embryos. Intermediately, the presence of green and purple embryos in the red/blue LED was also observed (Figure 2).

The use of darkness for the germination of *M. zehntneri* seeds drastically reduced the germination rate from 63.75% (control using light) to 11.3, 2.5, and 3.8% when seeds were cultivated in darkness for 10, 20, and 30 days, respectively (Table 1). Interestingly, seeds subjected to dark conditions for germination under short-period treatments, followed by exposure to light conditions, were not able to germinate for up to 12 months.

### 2.2. Germination of M. zehntneri Seeds In Vitro under Different Concentrations of Plant Growth Regulators

The germination percentage of *M. zehntneri* seeds using in vitro conditions in a culture medium was 74% (Figure 3), higher than observed using the same light conditions in the previous experiment with Petri dishes and filter paper (61.3%). Seed imbibition in an ethephon solution as a pre-treatment resulted in a significant and striking increase in the percentage of in vitro germinated seeds of *M. zehntneri*. The best treatment and response of seeds to ethephon was reported in the culture medium without phytoregulators, which resulted in 98% seeds germinated (Figure 3). In the control culture medium without the pre-treatment with ethephon, the germination rate was only 76% (Figure 3). The pre-treatment of seeds with ethephon also promoted faster seed germination, with the greater number of seeds germinated within 11–14 days, compared to the control (18 days) (Figure 4) and the highest GSI (2.93) compared to untreated (2.14) and the other plant growth regulators (BAP, GA_3_ or BAP + GA_3_) added to the culture media (Figure 5).

## 3. Discussion

### 3.1. Seed Dormancy in Cactaceae

Different authors reported dormancy in many Cactaceae species related to the xeric environment, with highly limited natural resources to support seed germination, seedling development, and survival. Some of these studies associate physical dormancy with mechanical resistance [16,17] caused by the rigid coat surrounding the embryo, which could limit the water uptake into the seeds. Seeds of different species of Cactaceae have rigid and hard coats [18,19]. However, recent studies increased the evidence that only physiological dormancy exists in Cactaceae seeds [20,21]. In *Melocactus*, the presence of seed dormancy was widely reported, with percentages of germinated seeds between 8 and 65%, depending on the species, genotype, harvest time, germination temperature, cultivation conditions (in vitro, Petri dish, or in vivo), and pre-treatments or treatments applied to the seeds [22,23,24]. However, the main biological evidence and reasons why 35–90% of seeds did not germinate, and the treatments required to further increase the germination of dormant seeds in Cactaceae, and more specifically in *Melocactus*, are not yet fully elucidated.

### 3.2. Light Strongly Influenced Seed Germination in Melocactus zehntneri

Regarding the wavelengths, except for monochromatic blue light that reduced the germination percentage of *M. zehntneri* seeds (38%), all other light wavelengths resulted in germination percentages between 61.0% and 62.0% under filter paper conditions. Except for monochromatic blue, all other LEDs tested in this study contained in their spectral composition, emission peaks in the red wavelength range, such as the monochromatic red, red/blue, and white LEDs. Red light appears to be the most important wavelength that promotes, increases, and accelerates the germination of *M. zehntneri* seeds. Red light is an important wavelength in mediating seed germination. The photomorphogenic responses, which involve light receptors like phytochromes, control plant development through the presence or absence of light and also the information and interpretation of different wavelengths in the environment, which guide the most appropriate development, including seed germination [25,26]. Cho et al. [27] also demonstrated that germination of *Arabidopsis* seeds is promoted in red light-enriched environments by the activation of Phytochrome B, resulting in a gradual increase in gibberellins that trigger germination. Interestingly, different studies in Cactaceae species sought to correlate the presence of light with increasing GA_3_ in seeds due to germination gains with these treatments [28]. However, in the present study, there was no strong evidence and no effectiveness of the application of external GA_3_. These results are in agreement with Barrios et al. [20] who analyzed different studies and did not find strong evidence of an association between GA_3_ and dormancy releasing in Cactaceae seeds [20]. The percentage of germinated seeds under different treatments rarely exceeds the average values observed in the present study with *M. zehntneri*.

The monochromatic blue light substantially reduced seed germination in *M. zehntneri*. The reduction in seed germination in response to blue light is not exclusive to the family Cactaceae, and it is frequently reported to inhibit the germination of dormant seeds in cultivated grasses [29,30]. Some studies suggest that the interaction of blue light with cryptochrome 1 results in an increased concentration of Abscisic Acid (ABA) that inhibits germination [31,32].

The majority (80.9%) from 275 studied Cactaceae species are considered positive photoblastic and, any negative photoblastic was reported [20,21]. Flores et al. [33] reported a strong influence of different Cactaceae tribes (phylogeny) on the response to light for seed germination. This study showed that the most of the Cactaceae tribes have a strong response to light for germination and that this differential response are due the size and weight of the seeds presented in different tribes [20,33]. Flores et al. [34] reported that from a total of 28 cactus species, all were considered positive photoblastic, and these authors also described the occurrence of secondary dormancy as a consequence of seed exposure to a period of darkness during germination. The same result was observed with *M. zehntneri* in the present study, in which light was necessary for seed germination, and exposure of seeds to dark conditions, even for short periods (10, 20, and 30 DAS), significantly reduced the seed germination percentage (Table 1). Interestingly, even after a short period of darkness (10 days), seeds did not germinate even after up to 12 months under the same light conditions that promote germination, with the probable acquisition of secondary dormancy due to the absence of light. Under light conditions, the percentage of germinated seeds of *M. zehntneri* was, on average, 40–60% [15], showing their photoblastic positive response.

### 3.3. Ethephon Releases the Germination of Dormant Seeds of M. zehntneri

Interestingly, the in vitro seeding and germination increased the germination percentage of seeds (76% in control) of *M. zehntneri* compared to the previous experiments in filter paper (≡60% germination) and showed that in vitro conditions can be used for this cactus species. The main differences between these two experiments explain that these differences are the result of the use of a culture media (nutrients, sucrose, and other components) instead of deionized water under filter paper conditions, as well as the use of pre-treatment with water for 24 h before seeding instead of only 10 min, which was used for the experiment with filter paper and previously reported in another study [15].

In our study, we used the in vitro conditions to test the effect of phytoregulators on the germination of this cactus. Treatments with phytoregulators have been used to break dormancy in seeds of different Cactaceae species. The most important effects observed in the present study with *M. zehntneri* occurred when seeds under in vitro conditions were pre-treated with ethephon (2-chloroethylphosphonic acid), showing a germination percentage of 98% of seeds, compared to 76% with water pre-treatment used as a control. In this way, ethephon was the unique treatment promoting the germination of dormant seeds in *M. zehntneri*, but the knowledge about how this phytoregulator affects cactus development is still limited. Among the few reports, ethylene is responsible for the closing, wilting, and pollination of flowers [35] and fruit ripening [36] in some cactus species. However, the effects of ethylene on releasing seed dormancy in Cactaceae have not yet been reported. Although little explored and used in Cactaceae, ethylene is considered a key hormone that regulates dormancy and seed germination, as well as the establishment of seedlings after germination in many plant species. This hormone is effective, at concentrations from 0.1 to 200 µL L^−1^, in releasing seeds from dormancy [13]. In the present study, the pre-treatment with ethephon was effective in releasing the germination of *M. zehntneri* seeds when applied by immersing seeds in a solution at 100 µL L^−1^ for 24 h using the commercial product Ethrel^®^ (Bayer^®^, Belford Roxo, Brasil), which contains 240 g L^−1^ ethephon, thus, at a concentration of 24 µL L^−1^ ethephon. Ethylene acts in the release of seeds from dormancy, especially involving a complex interaction with other hormonal groups in seeds, such as ABA, Gibberellins, Nitrous Oxide (NO), and reactive oxygen species (ROS) [37]. Different studies have demonstrated that, through the use of ethylene biosynthesis and action inhibitors, ET insensitive mutants, and the use of ET biosynthesis precursors, such as 1-aminocyclopropane 1-carboxylic acid (ACC), ethylene is involved in overcoming the dormancy and promoting germination [13]. In addition, this hormone can also neutralize the effects of ABA on seed dormancy.

Among the phytoregulators used to study germination, one of the most used was the GA_3_ [38]. This phytoregulator has shown divergent results in releasing dormant seeds of Cactaceae, in some cases even reducing the percentage of germinated seeds, as observed in *Ferocactus* species [39]. In the present study, seeds of *M. zehntneri* did not respond to the presence of GA_3_ at 1.0 mg L^−1^ in the culture medium (72% germinated seeds) and did not differ from the control without phytoregulators (76% germinated seeds). These results diverge from the in vitro germination of *M. sergipensis*, in which the addition of GA_3_ at 2 mg L^−1^ for 6 h increased the percentage of seeds germinated in this species. However, the maximum percentage of in vitro germinated seeds observed for this species was 38% [22], much lower than observed in our study with *M. zehntneri*. Nevertheless, the limited positive results in increased germination by using GA_3_ in several cacti species may be related to the differential sensitivity of seed tissues to GA_3_ [40]. 

Cytokinins and ethylene have been identified as hormonal groups that upregulate themselves and help to overcome seed dormancy [41]. However, in the present study, the addition of BAP, a synthetic cytokinin, or the BAP + GA_3_ combination in the culture medium reduced the effects associated with germination promoted by the pre-treatment containing ethephon. For example, in treatments containing this cytokinin in the culture medium, there were no differences in the percentage of germinated seeds pre-treated or not with ethephon. The effects of cytokinins on the germination of Cactaceae seeds are practically non-existent but positive effects of cytokinins on increasing the germination percentage and GSI have been demonstrated in species from other plant families [42,43]. 

## 4. Material and Methods

### 4.1. Plant Material

For the experiment, *Melocactus zehntneri* was used, from the germplasm collection of the Center of Agrarian Sciences, UFSCar, catalog number ABBC280 (Figure 6) by the Sistema Nacional de Gestão do Patrimônio Genético e Conhecimento Tradicional Associado (SisGen/Brazil). Three adult plants in full fruiting were used as a source of seeds, and fruit was collected at the moment they were detached from the cephalium (Figure 6).

### 4.2. Seed Preparation and Storage

Seeds were removed from the fruit and washed in a sieve under running water containing a few drops of neutral detergent to remove excess mucilage surrounding the seeds. After washing, seeds were placed in a grow room at 25–28 °C to dry on filter paper for 24 h. Seeds remained stored that way for another 14 days until the experiments. This procedure of removing mucilage and drying, demonstrated by previous experiments, is the best condition for the germination of seeds of this species.

### 4.3. Experiment under Different Light Intensities and Wavelengths for Germination

In the implementation of the experiment, seeds of *M. zehntneri* were immersed in distilled water for 10 min, according to results previously obtained by our laboratory team [14]. Later, using a laminar flow cabinet, seeds were subjected to asepsis, aiming at the reduction or even elimination of microorganisms. This was performed in 15 mL Falcon^®^ tubes and seeds were immersed in 70% alcohol (*v*/*v*) for one minute and then in a solution containing 30% sodium hypochlorite (2.0–2.5% active chlorine) added with 5 drops of neutral detergent for every 100 mL solution for 15 min under constant stirring, followed by three washes in previously autoclaved deionized water. In the last rinse, approximately 2 mL of deionized water with pH adjusted to 5.8 was kept as a vehicle for seed inoculation on the plates. All seeds, 20 per plate, were sown in clear, smooth, and sterile polystyrene Petri dishes, previously filled with two layers of filter paper saturated with 5 mL sterile deionized water at pH 5.8 per dish. This germination system was called filter paper condition. Subsequently, dishes containing the seeds were arranged under different light intensities (I, II, and III) and wavelengths given by LED lamps and cultivated in a grow room, as follows: blue LED (Phillips Greenpower LED Research Module Blue, ~440 nm, Pila, Poland), red LED (Phillips Greenpower LED module HF deep red, ~660 nm, Pila, Poland), blue (1)/red (1.5) LED (LabPar, with wavelength peaks at 447 nm, range 420–470 nm—blue and 667 nm, range 625–680 nm—red, Barueri, Brazil), and as a white LED control (Ourolux^®^, São Paulo, Brazil), with peaks at 440–450 nm (blue), 540–550 nm (green) and 610–620 nm (red).

Photosynthetically photon flux densities (PPFD) were measured using a PPFD Quantum meter, Apogee Instruments^®^, Model SQ-520 (North Logan, UT, USA) of each light source and were obtained by placing the dishes in equidistant distances from the LEDs (Table 2).

The experiment was a completely randomized design in a 3 × 4 factorial arrangement (PPFD × wavelength) with four replications composed of individual Petri dishes containing 20 seeds each. As a complementary experiment, dark time influence on the germination of *M. zehntneri* seeds was tested, in which seeds were kept protected from light in a grow room for three periods: 10, 20, or 30 days of darkness. In this case, seeds under germination conditions were only exposed to a light source after remaining in the respective periods in the dark. For this, the same procedure of preparation and inoculation of seeds was carried out, and the experimental design was completely randomized. Petri dishes from both experiments were sealed with a transparent PVC film and kept in a grow room at 26 ± 1 °C and a photoperiod of 14 h. Germination was evaluated twice a week. Seeds were considered germinated when the hypocotyl-radicle protrusion was equal to or greater than 0.1 cm. In the end, the Germination Percentage (G%), Average Germination Speed (AGS), and Germination Speed Index (GSI) were calculated.

### 4.4. In Vitro Germination of M. zehntneri under Different Concentrations of Phytoregulators

The main objectives of this experiment were to evaluate the effect of different classes of phytoregulators on the germination of *M. zehntneri* seeds and to develop a methodology for in vitro germination as an alternative to the method of seed germination in Petri dishes.

Procedures before seeding and aiming at seed asepsis were carried out as described in the previous experiment. In the last rinse, approximately 2 mL deionized water (pH~5.8) was maintained so that this solution containing the seeds could be used as a vehicle for inoculation in culture flasks containing 30 mL MS culture medium [44], containing sucrose (20 g L^−1^), inositol (100 mg L^−1^), activated charcoal (1 g L^−1^), and the pH was adjusted to 5.8 before the addition of agar (6.4 g L^−1^). Flasks containing the culture medium were autoclaved at 120 °C for 25 min. Phytoregulators tested for germination of *M. zehntneri* seeds and added to the MS culture medium were 6-benzylaminopurine (BAP) 4.44 µM; gibberellic acid (GA_3_) 2.89 µM; and the combination of the two, BAP (4.44 µM) add GA_3_ (2.89 µM), and control without addition of phytoregulators. Also, we evaluated the effect of pre-treatment of seeds in a solution containing 100 μL L^−1^ Ethrel^®^ (240 g/L ethephon—Bayer^®^, Belford Roxo, Brazil, totaling a concentration of 166 µM) for 24 h before the experiment and later inoculated in culture media containing the same treatments described above. After inoculation, flasks containing the seeds were kept in a grow room under the same conditions as in the previous experiment but using only the LED in the blue (1) and red (1.5) wavelengths.

The experiment was conducted in a Completely Randomized Design (CRD), in a 2 (pre-treatment with Ethrel^®^) × 4 (phytoregulators in the culture medium) factorial arrangement. In total, six replications were performed per treatment; each replication consisted of a flask containing 30 mL culture medium, with ten seeds per flask. Germination was checked twice a week, and seeds were only considered germinated when embryo protrusion was equal to or greater than 0.1 cm. After 42 days, the germination percentage (%G), the AGS, and GSI were also evaluated. Germination Percentage (G%) was determined by G% = ∑_ni/N × 100, where ∑ni is the total number of germinated seeds, and N is the total number of seeds tested [45]. The Average Germination Speed was determined according to the expression AGS = (ni)/ti, in which “ni” is the number of seeds germinated at time “i” and “ti” is the time after implementation of the test [46]. The Germination Speed Index (GSI) was determined using the equation GSI = G_1/N_1 + G_2/N_2 … + G_n/N_n, where G1, G2, … Gn refer to the number of germinated seeds and N1, N2, Nn, to the number of days after seeding [47].

### 4.5. Statistical Analysis

Data obtained were tested by analysis of variance (ANOVA). For the comparison of means, Tukey’s test was applied at 5% significance, using the software AgroEstat v1.1 [48] and RStudio v.3.1.1 [49].

## 5. Conclusions

An effective protocol was developed, which achieved high percentages of germination in *M. zehntneri*. Red light promoted and accelerated the germination of seeds of *M. zehntneri*, while monochromatic blue light reduced the germination percentage in this species. The increase of PPFDs of light is positively correlated with the germination of cactus seeds. The dark conditions during germination for short periods drastically reduced seed germination. Ethephon used as pre-treatment played a key role in the *M. zehntneri* seed germination (98%), releasing the germination of dormant seeds. The present study can serve as a reference for studies aiming to overcome seed dormancy in *Melocactus*, and in vitro germination can still serve as a tool for studying the germination and propagation of cacti species, aiming at their large-scale production or even for conservation purposes. Studies are required to increase knowledge of the effects of ethylene on the germination of seeds of other cactus species, as well as its mode of action in overcoming dormancy in Cactaceae.

## Figures and Tables

**Figure 1 plants-12-04127-f001:**
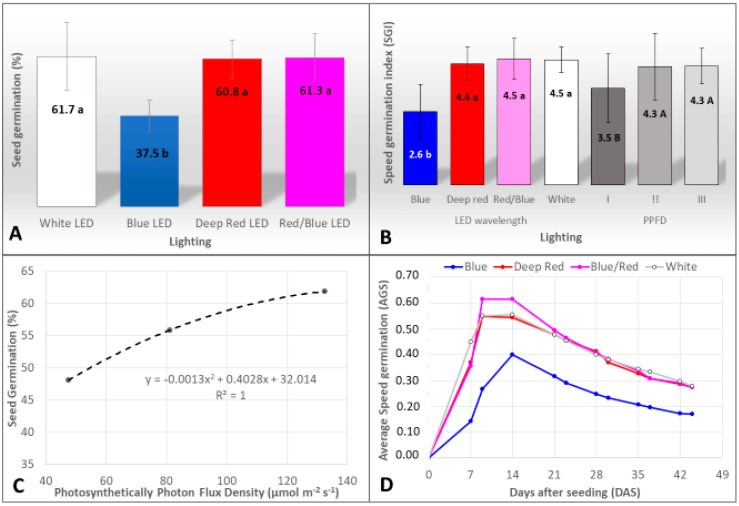
Germination (%) of seeds of *Melocactus zehntneri* cultivated under different light wavelengths and PPFDs sourced by LEDs: percentage of seed germination (**A**) (Different lowercase letters in the same column indicate the effects of the wavelength and uppercase letters indicate the effects of PPFDs on germination. Shapiro–Wilk Normality Test = 0.36, Levene test for homoscedasticity = 0.71. Coefficient of variation = 13.2%. F-values for: wavelengths (10.81 **), PPFDs (4.86 *), and Interaction (1.67 ns). Significant at 5% (*) and 1% (**) probability. ns, non-significant); germination speed index (GSI) (**B**) (Shapiro-Wilk Normality Test = 0.50, Levene test for homoscedasticity = 0.68. Coefficient of variation = 24.95%. F-values for: wavelengths (12.87 **), PPFDs (4.42 *), and interaction (2.05 ns). Significant at 5% (*) and 1% (**) probability. ns, non-significant); correlation between the percentage of seed germination and PPFD of light (**C**) and average germination speed under different LED wavelengths (**D**).

**Figure 2 plants-12-04127-f002:**
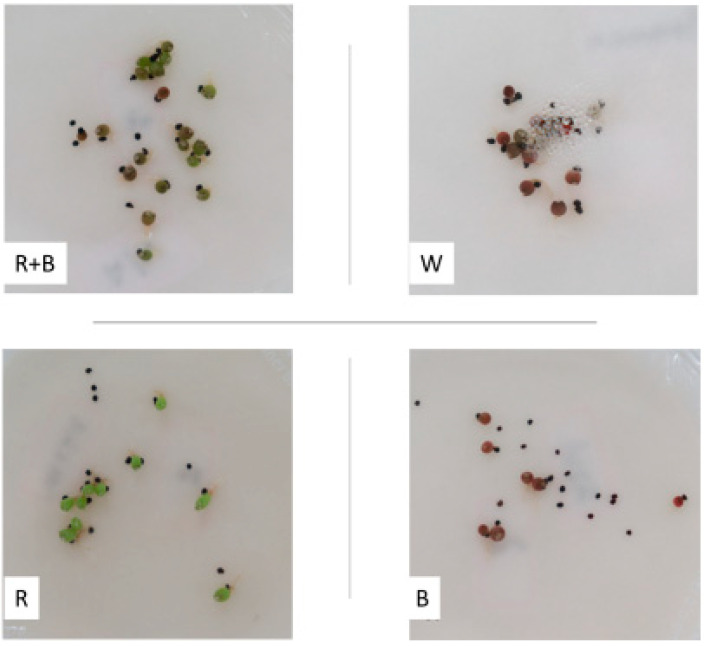
Seed germination and color of germinated embryos of *Melocactus zehntneri* cultivated under different light wavelengths: red and blue (R + B); white (W); red (R) and blue (B).

**Figure 3 plants-12-04127-f003:**
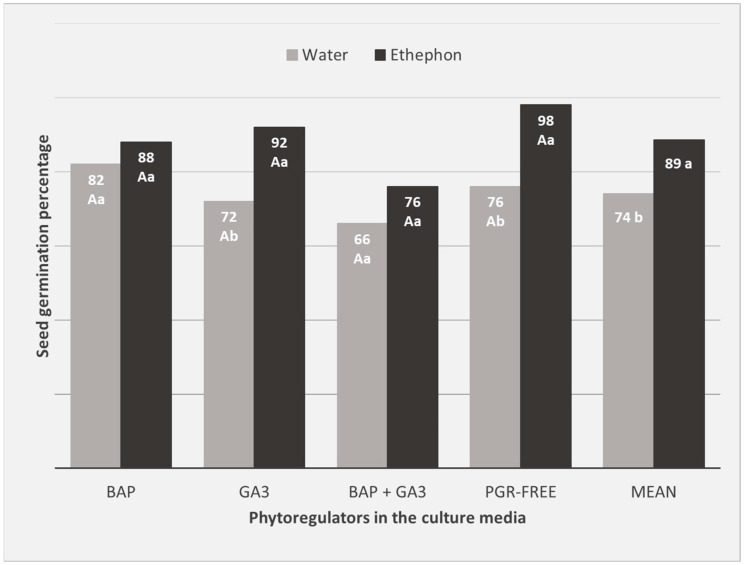
Germination percentage of *M. zehntneri* seeds subjected to the pre-treatment by soaking in ethephon solution or water (control) solution and cultivated under culture media containing the cytokinin 6-Benzylaminopurine (BAP) and/or Gibberellic Acid (GA_3_). Shapiro–Wilk normality test = 0.857; Levene test for homoscedasticity = 0.7278. Coefficient of variation = 32.7%. F-values for phytoregulators (1.58 ns), ethephon (11.37 **), and interaction (0.82 ns). Significant at 1% (**) probability. ns, non-significant. Uppercase letters indicate differences between the phytoregulators in the culture media and lowercase letters indicate differences between the pre-treatment of seeds with Ethephon or water. PGR-free + pre-treatment with water acts as a control.

**Figure 4 plants-12-04127-f004:**
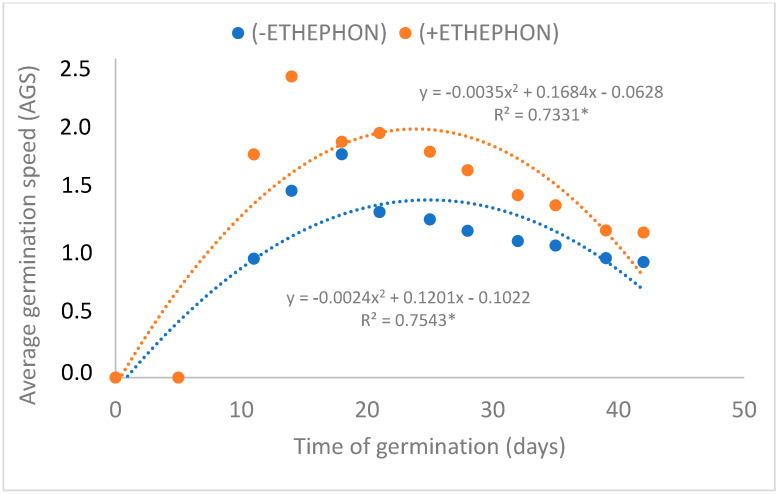
Average germination speed (AGS) of *M. zehntneri* seeds cultivated under different wavelengths (blue, deep red, red and blue, white) sourced by LEDs and photosynthetically photon flux densities (PPFDs). * Significant at 5% probability.

**Figure 5 plants-12-04127-f005:**
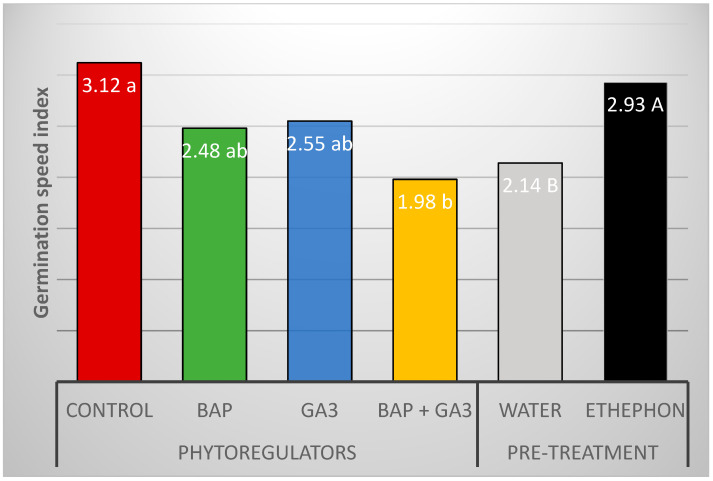
Germination Speed Index (GSI) of *M. zehntneri* seeds subjected to different phytoregulators added to the culture media and pre-treatment with ethephon. Shapiro–Wilk normality test = 0.63; Levene test for homoscedasticity = 0.0975. Coefficient of variation = 28.3%. F-values for phytoregulators (2.75 ns), ethephon (9.98 **), and interaction (0.49 ns). Significant at 1% (**) probability. Lowercase letters indicate differences between the phytoregulators in the culture media and uppercase letters indicate differences between the pre-treatment of seeds with Ethephon or water. ns, non-significant.

**Figure 6 plants-12-04127-f006:**
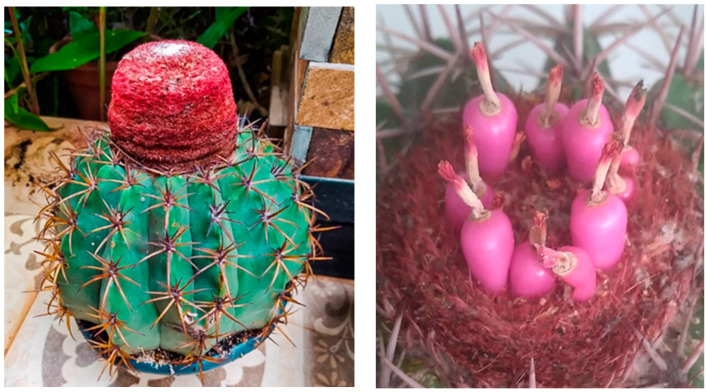
Adult plant of *M. zehntneri* with cephalium (**left**) and details of cephalium containing mature fruit (**right**) used to collect seeds for the experimental procedures. Original photos of JCC and MFCM.

**Table 1 plants-12-04127-t001:** Percentages of seed germination in *Melocactus zehntneri* cultivated under different dark times and measured after 45 days of seed maintenance under light conditions.

Dark Time	Percentage of Germination
Control under light conditions	63.75 a
10 days	11.25 b
20 days	2.5 b
30 days	3.8 b
F	31.386 *
CV(%)	84.223

* Means followed by different letters were significantly different by Tukey’s test at 5% probability. All germination data were evaluated after exposing seeds to light conditions for 45 days (control, 45 days; 10 days dark + 45 days under light; 20 days dark + 45 days under light, and 30 days dark + 45 days under light).

**Table 2 plants-12-04127-t002:** Photosynthetically Photon Flux Density (PPFD) sourced by LED wavelengths used for seed germination in *Melocactus zehntneri*. I, II, and III refer to the treatments using different light intensities where seeds were cultivated.

	Photosynthetically Photon Flux Density (µmol m^−2^ s^−1^)			
LED Wavelength	I	II	III	Mean
Blue	37.83 ± 0.50	67.19 ± 0.42	134.21 ± 0.54	79.74
Deep Red	53.59 ± 0.64	97.58 ± 0.73	144.89 ± 0.70	98.69
Blue/Red LED	41.28 ± 0.33	64.02 ± 0.75	115.19 ± 0.78	73.50
White LED	56.83 ± 0.59	96.19 ± 0.21	135.57 ± 0.36	96.20
Mean	47.38	81.25	132.47	

## Data Availability

The data could be shared if required.

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
