# Peer review of "Light and Ethephon Overcoming Seed Dormancy in Friar’s Crown (Melocactus zehntneri, Cactaceae), a Brazilian Cactus"

_plants, 2023, doi:10.3390/plants12244127_

Round 1
Reviewer 1 Report
Comments and Suggestions for Authors
Abstract
1. Awkward wording „that dormancy in Melocactus seeds may be associated with factors such as light and phytoregulators”
2. If 76% of the seeds germinated, it means that most of the seeds were not dormant.
Introduction
L.57-58 The % germination in vivo should be given; it is not known how many percent more germinated than in vitro
L.58-60 2. This is not true in paper NR14, 50% of the seeds germinated as a result of soaking in water, not as a result of scarification.
Results
L.70-72 “This section may be divided by subheadings. It should provide a concise and precise description of the experimental results, their interpretation, as well as the experimental conclusions that can be drawn.”- What's that supposed to mean? Any comments from a reviewer or adviser included in the work???
Table 1 should be in Material and Methods. What the numbers I, II, III mean in the table requires explanation.
“Figures, Tables and Schemes” - Why such a title in the Results section?
Table 2 – illegible
Fig.1 should be in Material and Methods
Fig.2 illegible
Fig.4 No scale on the vertical axis. Fig. poorly described, e.g. GA3 effect . There is no information that they were pretreated with water, such information appears in Fig. 6
Sometimes the term plant growth regulators is used, sometimes phytoregulators
M&M
Information about Germination Speed (AGS) and Germination Speed Index (GSI) should not be included in the statistical analysis.
Hormone concentrations should be given in moles.
Why were the seeds "immersed in water" first and only then treated with alcohol?
Discussion
The discussion was too long-winded, there was a lot of unnecessary information. The conclusions are partly too hasty.
Whether the germination of seeds was checked immediately after harvest?. Maybe 14 days of storage resulted in the release of dormancy if it occurs.
Does incubation in the dark really induce secondary dormancy? Maybe the seeds have lost their viability.
Are you sure your seeds are dormant, maybe they require light to germinate?
​I just posted some comments. In general, I think that the work is poorly written, contrary to the applicable rules for writing scientific works.
Author Response
Response to reviewers
Reviewer 1
Dear Reviewer, thanks for our efforts in revise our manuscript. We try to provide all of the suggestions you mentioned in your review, and we hope with these changes you can see the potential of the paper for publication in Plants. Best regards, Jean
Comments and Suggestions for Authors
Abstract
- Awkward wording „that dormancy in Melocactus seeds may be associated with factors such as light and phytoregulators”
Response: We changed this sentence to provide better meaning.
- If 76% of the seeds germinated, it means that most of the seeds were not dormant.
Response: We changed some sentences to provide better meaning of the study. In addition, the statistical analysis showed differences between the control pre-treated with water (76%) and the pre-treatment with ethephon (98%).
Introduction
L.57-58 The % germination in vivo should be given; it is not known how many percent more germinated than in vitro
Response: information provided
L.58-60 2. This is not true in paper NR14, 50% of the seeds germinated as a result of soaking in water, not as a result of scarification.
Response: This information was corrected in the sentence, thanks for your efforts.
Results
L.70-72 “This section may be divided by subheadings. It should provide a concise and precise description of the experimental results, their interpretation, as well as the experimental conclusions that can be drawn.”- What's that supposed to mean? Any comments from a reviewer or adviser included in the work???
Response: Thanks for your commentary. Yes, we deleted this part due this commentary from the editor were deleted and corrected in general in the text
Table 1 should be in Material and Methods. What the numbers I, II, III mean in the table requires explanation.
Response: The table 1 was transferred to MM section, and the light intensities I, II and III was clarified in the title of the table.
“Figures, Tables and Schemes” - Why such a title in the Results section?
Response: We used the template of the journal, that contains a subitem only for figures, tables and schemes
Table 2 – illegible
Response: The table were completely reformulated
Fig.1 should be in Material and Methods
Response: OK
Fig.2 illegible
Response: Maybe in the conversion for PDF the image reduces the quality. We try to increase the quality of this figure. Also, the title of figure was changed for better meaning
Fig.4 No scale on the vertical axis. Fig. poorly described, e.g. GA3 effect . There is no information that they were pretreated with water, such information appears in Fig. 6
Response: In the figure 4 we used a graphic type containing the values in percentage inside in the column. The title of this figure was rewriting and information requested were added. The information about water pre-treatment (control) were added.
Sometimes the term plant growth regulators is used, sometimes phytoregulators
Response: We standardized the term in the text
M&M
Information about Germination Speed (AGS) and Germination Speed Index (GSI) should not be included in the statistical analysis.
Response: deleted from this part and transferred to previous subsection
Hormone concentrations should be given in moles.
Response: We changed this information for µM.
Why were the seeds "immersed in water" first and only then treated with alcohol?
Response: the treatments with alcohol and sodium hypochlorite were used to eliminate superficial microorganisms with potential of contamination of culture media. But after these treatments, the seeds were washed for three times using distilled autoclaved water to eliminate these components from the seeds.
Discussion
The discussion was too long-winded, there was a lot of unnecessary information. The conclusions are partly too hasty.
Response: The discussion section was completely revised and shortened as recommended.
Whether the germination of seeds was checked immediately after harvest?. Maybe 14 days of storage resulted in the release of dormancy if it occurs.
Response: Yes, in the previous study we used seeds after harvest for seed germination, but the maximum seed germination obtained was 55%, when the aryl was removed from the seeds. With the aryl, no germination was observed.
Does incubation in the dark really induce secondary dormancy? Maybe the seeds have lost their viability.
Response: See that the experiment with dark conditions were realized at the same time with the tests of light and the seeds were from the same batch. Thus, we believe that seeds were induced to secondary dormancy, similar to other species. However we understood your position and rewritten the conclusions with care to avoid misinterpretation of the results. Thanks.
Are you sure your seeds are dormant, maybe they require light to germinate?
Response: All the experiments, under petri dishes that tested intensity and quality of light, and under in vitro conditions, were realized under light, and since under high intensity don’t germinated, demonstrated by the experiment 1. In the experiment 2, all seeds were submitted to light conditions that promote germination, but only submitted to ethephon resulted in highest germination percentages (98%). The results obtained in this manuscript and the previous paper published with the same species resulted in two year of experiments and all of them showed similar results.
​
I just posted some comments. In general, I think that the work is poorly written, contrary to the applicable rules for writing scientific works.
Response: Thanks for your negative appointments. We try to improve according your and all reviewer suggestions and hope you are able to see also the positive results contained in our manuscript.
Reviewer 2 Report
Comments and Suggestions for Authors
1- Please add the English name of the plant in title and scientific name in the parentheses.
2- References in the text should be written according to the format of journal, not on the basis of Superscript. Please, revise all the references in the text.
3- Please, reduce the number of paragraphs. There too many paragraphs in the manuscript, authors shall decrease them, and start each paragraph with new point and content.
4- Its seems the format of tables are not on the basis of journal s format.
5- until should be write the same and in the correct format. Please, check lines 260 and 262.
6- The Conclusion part is too short. Authors should revise the conclusion completely.
7- None of the references are on the basis of journal s format.
8- DOI should be added to all references
Author Response
Reviewer 2
Dear reviewer, thanks for your efforts in review our manuscript. Certainly that your contributions serves to improve our manuscript. Best regards, Jean Cardoso
Comments and Suggestions for Authors
1- Please add the English name of the plant in title and scientific name in the parentheses.
Response: There are no true translate for ‘coroa-de-frade’, thus we add the common name in Portuguese and improve the title for better meaning. See if ok.
2- References in the text should be written according to the format of journal, not on the basis of Superscript. Please, revise all the references in the text.
Response: The references in the text were formatted according the journal standards.
3- Please, reduce the number of paragraphs. There too many paragraphs in the manuscript, authors shall decrease them, and start each paragraph with new point and content.
Response: We were reduced the quantity of text, especially in the discussion section and reorganize the paragraphs according your suggestion, joining some paragraphs in only one according the subject.
4- Its seems the format of tables are not on the basis of journal s format.
Response: The tables were formatted according journal instructions
5- until should be write the same and in the correct format. Please, check lines 260 and 262.
Response: We checked these lines, but I not understood what we can do in these lines, sorry!
6- The Conclusion part is too short. Authors should revise the conclusion completely.
Response: The conclusion is completely revised and rewritten for better meaning
7- None of the references are on the basis of journal s format.
Response: All the references were revised and put in the format of the journal Plants
8- DOI should be added to all references
Response: The doi for references were didn’t added to maintain the standard format of the journal, according instructions for authors.
Reviewer 3 Report
Comments and Suggestions for Authors
Comments on the Quality of English LanguageMINOR
Author Response
Reviewer 3
Dear reviewer 3, thanks for your careful revision and commentaries that improved our text substantially. We hope we can get your message and corrected the main weaks you observed. Best regards, Jean
The draft “Light and ethephon overcoming seed dormancy in Melocactus 2 zehntneri (Cactaceae)” by Magnani and Cardoso have been reviewed. This draft refers to the study of the germination of Melocactus zehntneri (Cactaceae), a species abundant in Brazil and surrounding countries and which has not yet appeared in journals of any impact, as demonstrated in the first fifteen bibliographic citations. Very weak results and adequate discussion.
Minor. Line 53, gibberellic acid - OK; line 57,67 and 110, in vitro OK. Lines 71-73, delete OK. Tab. 2: rewrite to make it understandable. OK Fig. 4: what does the capital A mean?? OK What is the control in this figure?? OK Line 140 ,155, 160 cultivated,culture, cultivated OK. In discussion, ethephon not Ethephon OK; μL L-1 not μL L-1 OK; was Gibberellic Acid or GA3 OK;
Major.
The figures must be numerically arranged in Results.
Response: some figures were changed the position to attend the other reviewers. See if is ok now, thanks.
Line 79, “there was no interaction between these two factors”. Where do you get this conclusion??
Response: we improved the figure, as well as, revised this sentence for improve the meaning. See that the experiment under different light PPFD and wavelength, the both factors have significative differences, but not the interaction between ones (wavelength x PPFD), which means that independent of the wavelength the increase of PPFD are significative.
What means “light-irreversible dormancy”??
Response: we rewriting this sentence and deleted the part of ‘light-irreversible dormancy’ in results section and make this on the discussion section (Lines 217-228).
Given the poor description of Table 2, I cannot judge the results of this one (lines 98-103).
Response: we changed this figure for better description. The position of this table (now table 1) in the manuscript are altered to attend another reviewer. In addition, we improved the title information requested for better meaning.
However, the absence of germination in the dark seems interesting and should be properly commented.
Response: we improved these information in discussion section lines 217-228 and in conclusion section (lines 415-417)
Lines 106-108, I would like to know why you call the germination experiments in vitro and in vivo. Also in lines 14-17 (Summary). Line 191, cultivation conditions (in vitro, Petri dish or in vivo). Explain the differences.
Response: We corrected and add this information in different sections, and called filter paper conditions to differentiate to in vitro conditions.
Sec. 2.2, I hope that the experiments with Ethephon will be adequately commented in the discussion. And it’s.
Response: We improved this part and efforts to demonstrate the main effects of ethephon or ethylene in the germination of this cactus species (lines 236-275)
For me, this draft may be suitable for publication once minor and major
weaknesses are point by point solved.
The Referee
Round 2
Reviewer 1 Report
Comments and Suggestions for Authors
I wanted my comments to help the authors in their work. Progress is noticeable, the authors took advantage of my comments. However, Table 1 is still unreadable - Was germination in the light checked after incubation for various incubation periods in the dark? This should be clear from the title and description of the results.
I noted in the previous review that I am providing only some comments, counting on the authors to detect the shortcomings themselves. For example, the reader will not know after reading the “Introduction” and “Abstract” what phytoregulators were used.
The influence of phytoregulators except ethephon was not commented on in Abstract.
L.66-67 - "can solve and discover the main causes of dormancy" - this goal is missing in the work because the causes have not been explained or discovered. So it's better to leave these words out. I think the penultimate sentence of ”Conclusions” is to strong.
There could be many examples that require improvement, so I suggest the authors read the work carefully and thoroughly.
The Plants magazine is at level Q1, and the presented work is at level Q2. So I leave the decision about publishing to the Editor.
Author Response
Reviewer 1 -
I wanted my comments to help the authors in their work. Progress is noticeable, the authors took advantage of my comments. However, Table 1 is still unreadable - Was germination in the light checked after incubation for various incubation periods in the dark? This should be clear from the title and description of the results.
Response: The information requested are provided in the table. We hope that is according your expectation.
I noted in the previous review that I am providing only some comments, counting on the authors to detect the shortcomings themselves. For example, the reader will not know after reading the “Introduction” and “Abstract” what phytoregulators were used.
Response: We added this information in Abstract and Introduction section. Thanks for your support.
The influence of phytoregulators except ethephon was not commented on in Abstract.
Response: We added this information in Abstract
L.66-67 - "can solve and discover the main causes of dormancy" - this goal is missing in the work because the causes have not been explained or discovered. So it's better to leave these words out. I think the penultimate sentence of ”Conclusions” is to strong.
Response: The sentences were removed as recommended.
There could be many examples that require improvement, so I suggest the authors read the work carefully and thoroughly.
Response: We read carefully the paper and make some changes as requested.
The Plants magazine is at level Q1, and the presented work is at level Q2. So I leave the decision about publishing to the Editor.
Response: Thanks for your efforts in contribute with the increase the quality of our paper.
Reviewer 2 Report
Comments and Suggestions for Authors
(1) The English language in the manuscript needs Minor revision. The authors can ask one native English language speaker read it or they double-check the text for some grammatical errors.
(2) The English name of the plant should be written in Abstract followed by its Scientific name.
(3) in vitro germination should be deleted in keywords and replaced by another keyword. The keywords in the section of keywords should be written on the basis of alphabetic order.
(4) Authors have used many paragraphs in Introduction (and other parts of the manuscript), they can reduce the paragraphs, and each paragraph should be started with new point and subject.
(5) It seems the format of Tables are not on the basis of journal s format. Please, re-check them.
(6) Please, revise the conclusion and just write key points as well as your suggestions and recommendations for future researches. Also avoid using many paragraphs in Conclusion.
(7) DOI for all published articles in Reference section should be added if they have. The format of some references are not on the basis of Journal s format. Please, check them all.
(8) It is also recommended that authors add one part after conclusion and before references as Abbreviations, and add the words in this section.
(9) In some parts of the text in vitro is NOT Italics, they should be all written in Italics. All scientific names should be written in Italics.
Comments on the Quality of English LanguageThe English language of the manuscript needs Minor editing.
Author Response
Reviewer 2
(1) The English language in the manuscript needs Minor revision. The authors can ask one native English language speaker read it or they double-check the text for some grammatical errors.
Response: We used an editing service for English minor revisions for corrections of some grammatical errors, as requested.
(2) The English name of the plant should be written in Abstract followed by its Scientific name.
Response: This information was provided.
(3) in vitro germination should be deleted in keywords and replaced by another keyword. The keywords in the section of keywords should be written on the basis of alphabetic order.
Response: We replaced by only ‘germination’
(4) Authors have used many paragraphs in Introduction (and other parts of the manuscript), they can reduce the paragraphs, and each paragraph should be started with new point and subject.
Response: We reduced the number of paragraphs as recommended.
(5) It seems the format of Tables are not on the basis of journal s format. Please, re-check them.
Response: We did minor modifications in the tables to provide the journals format.
(6) Please, revise the conclusion and just write key points as well as your suggestions and recommendations for future researches. Also avoid using many paragraphs in Conclusion.
Response: We rewritten the conclusion section following your instructions.
(7) DOI for all published articles in Reference section should be added if they have. The format of some references are not on the basis of Journal s format. Please, check them all.
Response: The Doi of articles were added to the references and some references were formatted for journal instructions
(8) It is also recommended that authors add one part after conclusion and before references as Abbreviations, and add the words in this section.
Response: added a subsection called Abbreviations. All references were checked and corrected according the journal instructions
(9) In some parts of the text in vitro is NOT Italics, they should be all written in Italics. All scientific names should be written in Italics.
Response: The word ‘in vitro’ were written in italics and all scientific names were revised for maintain in italics
Round 3
Reviewer 1 Report
Comments and Suggestions for Authors
Now is Ok
Author Response
Thank you very much for your contributions and corrections. Best regards, Jean Cardoso